# Recent Advances in Ammonia Gas Sensors Based on Carbon Nanomaterials

**DOI:** 10.3390/mi12020186

**Published:** 2021-02-12

**Authors:** Alexander G. Bannov, Maxim V. Popov, Andrei E. Brester, Pavel B. Kurmashov

**Affiliations:** 1Department of Chemistry and Chemical Engineering, Novosibirsk State Technical University, 630073 Novosibirsk, Russia; m.popov@corp.nstu.ru (M.V.P.); brester94@gmail.com (A.E.B.); kurmashov@corp.nstu.ru (P.B.K.); 2Zelinsky Institute of Organic Chemistry, Russian Academy of Sciences, 119991 Moscow, Russia

**Keywords:** gas sensor, ammonia, graphene, carbon nanotubes, graphene oxide, response, chemiresistive sensor, response, sensitivity

## Abstract

This review paper is devoted to an extended analysis of ammonia gas sensors based on carbon nanomaterials. It provides a detailed comparison of various types of active materials used for the detection of ammonia, e.g., carbon nanotubes, carbon nanofibers, graphene, graphene oxide, and related materials. Different parameters that can affect the performance of chemiresistive gas sensors are discussed. The paper also gives a comparison of the sensing characteristics (response, response time, recovery time, operating temperature) of gas sensors based on carbon nanomaterials. The results of our tests on ammonia gas sensors using various techniques are analyzed. The problems related to the recovery of sensors using various approaches are also considered. Finally, the impact of relative humidity on the sensing behavior of carbon nanomaterials of various different natures was estimated.

## 1. Introduction

The detection of toxic and flammable gases is of interest in terms of environmental protection and labor protection. Gas sensors are usually used for the monitoring of their content in air [1,2,3,4]. The search for low-cost and effective materials for gas sensors is extremely crucial for the monitoring of concentrations of harmful gases in the air.

Ammonia is one of the most commonly used chemicals. The exposure limit of ammonia to humans is 35 ppm for 10 min according to Occupational Safety and Health Administration [5,6], showing that this substance is incredibly dangerous. It can be noted that the content of ammonia in the atmosphere is relatively low and only reaches the level of ppb (1–5 ppb) [2,7]. At the same time, this gas is extremely aggressive and highly toxic and has harmful effects on the skin, eyes, digestive tract, and mouth [8,9].

Ammonia gas sensors can be used in various fields of industry, e.g., chemical, petroleum and petrochemical, where the probability of accidents and the pollution of air with ammonia are very high and the determination of low concentrations is necessary. However, the direction of the research is extremely important for biomedical applications. Ammonia is one of the biomarkers used to estimate disease and can be successfully used in medical diagnostics by means of an analysis of exhaled breath [10]. It is the product of the metabolism of amino acids and is therefore present in relatively low concentrations [11]. Typically, exhaled air consists of 78.62% N_2_, 13.6% O_2_, 4–5% CO_2_, and 1% of other gases [12]. Commercial gas sensors have a range of operating concentrations (1–1000 ppm NH_3_) and a relatively high operating temperature (a few hundred degrees Celsius) [13], both of which are inappropriate for breath analysis. Therefore, there is a strong need to develop a room temperature gas sensor with a lower detection limit relative to ammonia with good selectivity, especially toward gases found in exhaled air. The range of concentration of ammonia for a healthy person is 0.5–2 ppm [14]. In [15,16], ammonia detection was used to assess halitosis. The correlation between the concentration of methyl mercaptan and ammonia produced by the bacteria derived from the tongue and dental plague was found in [15]. Abnormal concentrations of ammonia in exhaled air are mainly attributed to the malfunctioning of the kidneys [17]. Kearney et al. [18] used the analysis of ammonia in exhaled breath in order to detect an infection caused by the presence of *Helicobacter Pylori* using a fiber optic sensor. The positive subjects had lower concentrations of ammonia (0.04 ± 0.09 ppm) compared to negative subjects (0.49 ± 0.24 ppm), whereas patients that underwent treatment showed values of ammonia concentration that changed from positive to negative. One of the main advantages of the application of chemiresistive gas sensors for breath analysis is the correlation between the concentration of ammonia in breath and blood urea nitrogen [19], which makes the use of these sensors more convenient and simple for medical diagnostics.

There are two main methods for detection of ammonia: solid-state and optical methods. Solid-state methods relate to various types of sensors based on metal oxides, conducting polymers, electrochemical sensors, surface acoustic wave sensors, field effect transistor sensors, etc. These methods are relatively simple and suitable for portable devices. The use of optical methods such as gas chromatography coupled with mass spectrometry has an advantage as high accuracy, but they are time consuming, have complicate preparation techniques, and need qualified staff [20,21]. Chemiresistive gas sensors are easier and faster, which makes them attractive for detection of ammonia [22].

Nanotechnology makes it possible to bring new unique properties for materials, which is extremely important for the creation of advanced environmental sensors. Carbon nanomaterials constitute a wide class of popular materials that has attracted researchers all around the world, and the application of these nanomaterials in gas sensors is of significant interest [23,24,25,26]. Gas sensors based on carbon nanotubes were initially used for the detection of various dangerous gases, e.g., NO_2_ [27,28,29,30,31,32,33,34], NO [35,36,37], H_2_ [38,39,40,41,42], volatile organic compounds [43,44], BTX-type gases (benzene, toluene, xylene) [45,46], CH_4_ [47,48,49], CO [50], CO_2_ [51], H_2_S [52,53,54], chloromethanes [55], CS_2_ [56], aromatic hydrocarbons [57], formaldehyde [58,59,60], and ethanol [61]. Then, the research on graphene and related materials accelerated toward the application of these materials in gas sensing [26,62,63,64,65,66,67]. The creation of nanohybrid architecture sensors [68,69,70,71,72], combining carbon nanomaterials with various compounds and materials, made it possible to enhance the sensing behavior.

This review is devoted to assessing the performance characteristics of gas sensors based on carbon nanomaterials for the detection of ammonia, along with the problems related to their creation, design, and operation. The results regarding the creation of room temperature chemiresistive sensors based on carbon nanotubes, carbon nanofibers, graphene, graphene oxide, and related materials are presented.

## 2. Ammonia Gas Sensors and Gas Sensing Mechanism

The majority of publications on carbon nanomaterials are devoted to chemiresistive [73,74,75] gas sensors, although there are also chemicapacitive [76], surface acoustic wave [77], electrochemical [78], optic fiber [79], and quartz crystal microbalance [80] ammonia gas sensors (Figure 1). Each sensor based on carbon nanomaterials has its own mechanism of NH_3_ detection.

The chemicapacitive sensor shows the change of capacitance when exposing ammonia. The surface acoustic wave consisted of two interdigital transducers, a sensing layer, and reflectors. The mechanism of this sensor is based on the interaction between the gas molecules and the sensing layer on a piezoelectric substrate. This interaction causes perturbations on the boundary conditions of the propagating surface acoustic wave, manifesting as a change of velocity and attenuation of the propagating wave [81]. Kim et al. [76] reported that the strong electric field applied between a silicon substrate and single-walled carbon nanotubes (SWCNTs) was generated outside of the array of carbon nanotubes, causing the polarization of absorbed molecules of ammonia that induces the change of capacitance.

An electrochemical gas sensor consists of the working electrode and counter electrode with the electrolyte between them (sometimes there are three or four electrodes). The diffusion of gas into the sensor through the membrane induces the processes of reduction or oxidation on the working electrode, changing the current passing [78,82]. The fiber optic gas sensor operates using the measurement of optical absorption at certain wavelengths. The optical fiber transfers light from the absorption cell [83]. Yu et al. [79] investigated the platinum-nanoparticle-incorporated graphene oxide fiber optic sensor. The role of graphene oxide is the change of refractive index of microfiber coated with Pt/graphene oxide films. The index became more sensitive to the change of NH_3_ concentration using graphene oxide.

The deposition of the gas sensing layer on the quartz crystal microbalance (QCM) makes it possible to detect ammonia. Piezoelectric is placed between two electrodes. The resonant frequency changing during contact with gas changes proportionally the mass of the adsorbed layer [84]. The role of carbon nanotubes used for QCM is in the enhancement of absorption of ammonia molecules and the subsequent increase of mass detected by microbalances, which is especially higher when using functionalized carbon nanotubes [85].

Typically, the chemiresistive gas sensors undergo a change of resistance when exposed to ammonia. The existing chemiresistive gas sensors are based on semiconductors and operated at relatively high temperatures (above 250–300 °C). The use of carbon nanomaterials makes it possible to decrease their operating temperature to room temperature, which decreases their energy consumption, on the one hand, and reduces the risks of explosions on the other hand [41]. These sensors possess good selectivity and high response that is important for their further application and scale up [5]. Sensing layers of carbon nanomaterials can be created using various techniques: direct deposition on the sensor substrate [86], screen printing [87], spin coating [73], layer-by-layer deposition [72], Langmuir–Blodgett deposition [88], airbrushing [46], etc.

The integration of sensors into portable devices requires the creation of devices based on carbon nanomaterials with a high surface area, such as carbon nanotubes [23,89], graphene oxide [90,91], graphite oxide [92], reduced graphene oxide [93,94,95], nanoporous carbons [96,97,98], fluorinated graphene [99,100,101], aerographite [102,103], etc. Initially, the majority of the research was based on the creation of flexible films for the detection of aggressive gases, including ammonia. However, many researchers have turned to the investigation of more convenient materials, such as polyimide [104] and polyethylene terephthalate (PET) [24,105]. Nowadays, active materials for gas sensors are deposited in the form of films, which are very sensitive to any changes in gas composition. However, pellet sensors also exist and display a sufficient sensitivity to the vapors of solvents, such as acetone, chloroform, gasoline, and alcohol [25].

Despite the difference of various carbon nanomaterials, the mechanism of change of resistance when exposing ammonia is almost not changed among them. Most of the carbon nanomaterials undergo an increase of resistance during the adsorption of ammonia that is related to the p-type of these materials (e.g., multi-walled carbon nanotubes (MWCNTs), carbon nanofibers (CNFs)). Ammonia donates electrons to the active materials of the sensor, inducing a decrease of concentration of charge carriers (holes) [87,106]. The mechanism of change of hybrid sensor resistance is complex and depends on the type of hybrid. For example, in a polypyrrole (PPy)-reduced graphene oxide (rGO) sensor, both materials behave as sensing ones and enhance the capture of ammonia molecules, inducing an increase of resistance [63]. The conjugated π-systems of hybrids zinc(II) phthalocyanine–SWCNTs (both components possess p-type conductivity) led to a larger response [107]. Some experiments on the determination of gas-sensing behavior were carried out in UV light, which may change the mechanism. For example, in [108], pristine graphene showed the increase of resistance as the same effect for many carbon nanomaterials. Whereas the opposite behavior was found in UV light; i.e., there was a change of conduction state of graphene.

Pandey et al. [109] mentioned the typical features that a gas sensor should possess: (i) operation at room temperature; (ii) working in ambient environment and no requirement of oxygen or air supply; (iii) no external stimulus such as Joule heating or UV illumination for response/recovery; (iv) low detection limit; (v) high sensitivity and reproducibility; (vi) fast response and recovery; (vii) low cost and eco-friendly, etc. This list is extended enough, but some points may be added. The ideal gas sensor must possess the stabile response regardless of the relative humidity of air and the presence of any other gases lasting for a long time. Taking into account the results of an investigation of a carbon-based sensor at room temperature (20–30 °C), one can note that the ideal sensor could operate at real temperatures of outdoor air (at least from −10 to +40 °C). The range of the concentrations of ammonia to be detected ranges from a few ppb to hundreds of ppm. However, some of the authors already reported the sensors are capable of detecting a few ppt (parts per trillion), e.g., guar gum/Ag nanocomposite film [109]. The sensors based on carbon nanomaterials approach this level. In [108], the detection limit of a pristine graphene-based sensor was detected at the level of 33.2 ppt. Rigoni et al. [110,111] created the chemiresistive gas sensor based on pristine SWCNTs with a detection limit of 3 ppb. Therefore, the application of carbon nanomaterials should provide all these features to a new generation of ammonia gas sensors.

## 3. Gas Sensors Based on Carbon Nanotubes

Gas sensors based on single-walled carbon nanotubes are frequently investigated for ammonia detection. The first publications on ammonia gas sensors were devoted to SWCNTs, among other materials. The sensing behavior of pristine SWCNTs is relatively low; therefore, various approaches were used for its enhancement, such as the decoration of SWCNTs with semiconducting and metallic nanoparticles [112], treatment with acids [113], functionalization with conducting polymers [114], the creation of hybrids [115], etc.

Bekyarova et al. [116] reported on SWCNTs functionalized with *m*-aminobenzene sulfonic acid (PABS) deposited on interdigitated electrodes and used for the detection of ammonia, indicating the enhanced response compared to purified SWCNTs. The sensors showed a two times higher response compared to purified SWCNTs. The active layer was represented by the random network of SWCNTs with both semiconducting and metallic nanotubes, taking into account that the change in the resistance of metallic nanotubes did not bring significant changes in terms of sensor resistance. Sensors showed an incomplete recovery of N_2_ flow, and a longer period is necessary to carry out complete recovery. The mechanism of ammonia interaction with PABS-functionalized SWCNTs is the change in the electronic structure of PABS as a result of its deprotonation, which induced hole depletion and a reduction in the conductivity of the functionalized SWCNTs.

There are many articles devoted to the use of various carbon nanomaterials (single-walled carbon nanotubes [23], multi-walled carbon nanotubes [117], graphene [118], reduced graphene oxide [119], graphene nanoribbons [120,121]) for the enhancement of the gas-sensing properties of ammonia sensors based on polyaniline (PANI). Extended research on the creation of a sensing array for breath analysis based on SWCNTs functionalized with various semiconducting organic molecules was carried out by Freddi et al. [23] (Figure 2). The sensors were developed on plastic substrates. The highest response was achieved for PANI-functionalized single-walled carbon nanotubes, which are p-type semiconductors and significantly increase the response to reducing gas, e.g., ammonia. The enhancement of sensing characteristics was also achieved in [122] via a SWCNTs–OH/PANI composite. The positive effect of the addition of SWCNTs to PANI was attributed to the structure formed due to hydrogen bonds between carboxylated carbon nanotubes and PANI, increasing the probability of the interaction of ammonia molecules and the SWCNTs–OH/PANI composite.

Flexible films of SWCNTs functionalized with carboxylic groups were also used for ammonia detection in [123]. It was shown that the sensor showed a response of 30% to 300 ppm NH_3_ compared to 15% for pristine SWCNTs (the response was determined in nitrogen). The authors reported that the pristine carbon nanotubes showed the same resistance without recovery after contact with ammonia, whereas the functionalized film based on CNTs treated in aqua regia was recovered. Heating to 40 °C only led to the immediate return of the resistance to the values of the baseline. The enhanced response of functionalized SWCNTs was attributed to the formation of hydrogen bonds between ammonia molecules and oxygen and/or OH groups on the surface of nanotubes. This effect forms possible charge traps. However, the functionalization of SWCNTs with carboxyls cannot be considered as an effective method to improve the sensing characteristics. Rigoni et al. [24] achieved a more than 100-fold increase in ΔR/R_0_ for SWCNTs functionalized with CTAB (cetyltrimethyl ammonium bromide) surfactant compared to COOH–SWCNTs (a fraction of the groups was 1–3 at %) in the range of 10–30 ppm NH_3_. The response time to 10 ppm at relative humidity (RH) of 40% was almost the same for both SWCNTs (16.7 ± 0.2 s and 16.2 ± 0.2 s, respectively), but the recovery time was higher for COOH-SWCNTs (127.9 ± 2.1 s and 116.8 ± 1.5 s, respectively). The cause of this effect was not explained by the authors. The lowest concentration at which the sensitivity of COOH-SWCNTs was determined was 30 ppb (sensitivity S was ≈10 ppm^−1^), and most of the results published are not examined at this extremely low concentration of ammonia. The sensitivity of CTAB-SWCNTs was approximately two orders of magnitude lower compared to COOH-SWCNTs. Such a difference can be related to the feature of the sensing layer, where the CTAB layer hinders the interaction with ammonia, since CNT bundles are covered. In contrast, the layer functionalized with COOH groups is thinner and has a larger surface area; therefore, it is more susceptible to any change of concentration of NH_3_. In addition, the COOH-SWCNTs active layer showed a more stable sensor response in a range of 1–30 ppm. According to combined exposure of various relative humidity values (0–80% RH), this sensor is more stable compared to CTAB-SWCNTs.

The functionalization of SWCNTs with phthalocyanines made it possible to significantly improve sensing characteristics by means of the covalent or non-covalent functionalization of carbon nanotubes [108]. The cross-sensitivity data presented in [124] showing the significantly higher sensitivity of SWCNT/substituted silicon (IV) phthalocyanine sensors to volatile organic compounds (VOCs) and CO_2_ make them attractive for the analysis of exhaled breath. In [107], it was found that SWCNTs covalently functionalized with 1-[N-(2-ethoxyethyl)-4-pentynamide]-8(11),15(18),22(25)-tris-{2-[2-(2-ethoxyethoxy)ethoxy]-1-[2-((2-ethoxy ethoxy)-eth-oxy)methyl]ethyloxy}zinc(II) phthalocyanine (ZnPc) possessed higher responsiveness that non-covalently functionalized SWCNTs, which was in correlation with the number of ZnPc molecules adsorbed. It is interesting that the functionalization of reduced graphene oxide does not lead to the improvement of the sensor response, since the functionalization of rGO with ZnPc induced a reduction of active sites compared to pristine rGO. The creation of hybrid materials with phthalocyanine derivatives is successively achieved not only for SWCNTs but also for reduced graphene oxide (rGO) [125,126]. It can be noted that the hybrids rGO/3-CuPc [125], Kadem [126] showed higher response compared to pristine rGO.

Multi-walled carbon nanotubes (MWCNTs) are rarely used for gas-sensing applications compared to SWCNTs. Pristine MWCNTs possess a relatively low response [127,128]; therefore, various approaches to functionalization have been used. However, the problem with recovery arises after the functionalization of MWCNTs. Sharma et al. [127] reported on the complete recovery of pristine MWCNTs/Al_2_O_3_ composites after 24 h, whereas the sensor based on acid-treated MWCNTs/Al_2_O_3_ showed no recovery. This fact indicates the domination of the chemisorption of ammonia on the surface of acid-treated MWCNTs enriched with carboxyls and hydroxyls. The stronger interaction of ammonia with the surface of oxidized nanotubes than with pristine ones was confirmed by SCC-DFTB computations [113].

One of the ways to change the electronic properties of CNTs is their doping. However, the application of doped CNTs in ammonia detection is poorly studied. In [25], aligned nitrogen doped MWCNTs were synthesized by the decomposition of benzylamine/ferrocene solutions (850 °C, Ar atmosphere). It has been shown that the change in the resistance of the sensor is caused by the bonding of molecules of ammonia to pyridine-like sites. Other types of nitrogen-doped carbon nanomaterials, e.g., N-doped carbon spheres [129], activated carbon [130], SWCNTs [131], and graphene [132], have also shown their efficiency in the detection of ammonia. However, there are some differences in the effect of doping on the response. For example, Panes-Ruiz et al. [133] have shown that pristine SWCNTs had the same response as B-SWCNTs and N-SWCNTs. It was concluded that ammonia molecules interact with carboxyl groups on the sidewalls of SWCNTs rather than with the atoms of dopants. Therefore, the additional oxidation can enhance the sensing behavior rather than the doping of SWCNTs with boron or nitrogen.

The creation of various composites based on MWCNTs, a matrix of conducting polymers (e.g., poly(3,4-ethylenedioxythiophene)–polystyrene sulfonic acid [134], polyaniline [135,136], polythiothene [137], etc.), and metal oxides (TiO_2_ [138], Al_2_O_3_ [139]) made it possible to significantly improve the sensing behavior toward ammonia. Moreover, there are some hybrids used [140] where two types of carbon nanomaterials are applied, such as MWCNTs and reduced graphene oxide (rGO). It was reported that rGO–MWCNTs composites made it possible to improve the conductivity of polypyrrole, since these materials are both highly conductive. However, the role of MWCNTs in charge transfer and its contribution to the sensing characteristics cannot be clearly estimated. An important role of MWCNTs in composites with conducting polymers is the enhancement of the surface area of polymers, which increases the number of active sites for ammonia adsorption in turn [137].

Many authors have reported on the p-type behavior of MWCNTs, the resistance of which increases upon ammonia exposure [86,141]. However, there are some exceptions, where the resistance of MWCNT film undergoes a decrease during contact with ammonia [142].

Plasma treatment is successfully used to improve the sensing characteristics of ammonia gas sensors. Woo Ham et al. [143] carried out the modification of MWCNTs (Fe/Mo/Al_2_O_3_ catalyst, 923 K, C_2_H_4_, purified in 3 N HNO_3_) in oxygen plasma (30 sccm O_2_, 20 W RF power). The sensitivity of modified gas sensors toward ammonia was two times higher than that of untreated sensing material. Plasma treatment made it possible to increase the concentration of oxygen from 16.37 at % (pristine MWCNTs) to 34.09–49.32 at % (plasma treated MWCNTs). The enhancement of sensitivity was explained by the formation of hydrogen bonds between ammonia and oxygen-containing functional groups on the surface of MWCNTs that led to a decrease in the density of holes in carbon nanotubes. Kim et al. [144] created a micromachined ammonia gas sensor based on an O_2_-functionalized MWCNT/PANI sensor, reaching a response of 3.34%/(ppm NH_3_) within a range of 10–100 ppm NH_3_. The role of oxygen plasma was in the creation of sites for functional groups. In [69], PANI-coated MWCNTs (MWCNTs/33 wt % PANI) showed good stability and responsivity.

In [142], sensors based on plasma-treated MWCNTs were created. Plasma functionalization was carried out using oxygen plasma followed by plasma co-polymerization of maleic anhydride and acetylene. The oxygen plasma led to the formation of a core–shell carbon structure (Figure 3a) as a result of the etching of carbon nanotubes, and the second type of treatment induced the formation of functional groups on the surface of these structures (Figure 3b). The response of plasma-treated carbon structures reached 22.5% at 100 ppm compared to 7.1% for pristine MWCNTs (Figure 3c).

Another approach to enhance the sensing characteristics of gas sensors based on MWCNTs is the creation of a hybrid material in comparison to functionalization. Abdulla et al. [135] investigated an Ag polyaniline/MWCNT nanocomposite for breath analysis. The strips were obtained by spin coating. The application of sensors for breath analysis was analyzed in the presence of CO_2_. The stable behavior of the sensor was achieved at a relatively high concentration of carbon dioxide (above 40,000 ppm). Despite the impact of relative humidity, the sensor showed an unchanged response to ammonia under different levels of humidity (65–90% RH).

The creation of hybrids based on MWCNTs and metal oxide semiconductors makes it possible to enhance the sensitivity of gas sensors compared to pristine MWCNTs. In [138], catalytically grown MWCNTs were modified with TiO_2_ using atomic layer deposition, whereas the interface between carbon nanotubes and titanium dioxide was functionalized by carboxyl plasma polymers. The formation of a p–n heterojunction made it possible to achieve a response of ≈2.5% toward 100 ppm NH_3_, which depended on the operating temperature of the sensor and increased from 25 to 100–150 °C. The creation of such a heterojunction is frequently used in hybrid materials based on CNTs [145]. Le et al. [146] found that the composite based on MWCNTs and WO_3_ nanobricks showed a 28-fold higher response and a twofold lower recovery compared to the pristine MWCNT sensor. This synergetic effect can be explained by the fact that WO_3_ behaves as a p-type semiconductor at a low temperature, although it is an n-type semiconductor naturally. Such a huge increase in response in hybrids with metal oxides was also observed in [147] for CNT/ZnO nanowire networks, and this effect is attributed to the higher transfer of electrons achieved in hybrid materials upon exposure to ammonia. Recently, sensors based on green materials and biopolymers were created [148,149,150]; therefore, the study of hybrids of these materials and carbon nanomaterials is expected in the future.

The majority of publications consider the dominant role of active materials in the formation of sensing properties, whereas the contribution of the method of deposition is poorly considered. In [151], a comparison of the sensing characteristics of MWCNT–PEDOT:PSS sensors created using inkjet printing and drop casting was carried out, showing the increased response of the first technique. This is explained by the homogeneous gain effect.

The results regarding the stability of the sensing characteristics of CNT-based sensors are very important. Loghin et al. [152] carried out long-term stability tests of ammonia sensors based on CNTs (Figure 4). A retainment of up to 96% of the response after 1 year and 4 years of storage under ambient conditions was found.

Finally, we must note that the interconnection between specific properties of CNTs and their sensing behavior is still a matter of discussion. Taking into account the fact that the adsorption plays an important role in the capturing of ammonia molecules by carbon materials, it would be desirable to make a comparison of the surface area and other textural properties of the materials. Such research is complicated by the fact that the number of CNTs in the active layer of gas sensors is extremely low, and it is not possible to carry out this study from a technical point of view.

In our consideration of sensors based on CNTs, we should take into account that there are also sensors based on carbon nanofibers (CNFs) [104,153,154,155]. Since CNFs are cheaper than CNTs and semiconducting metal oxides, they become excellent candidates for sensing applications [154]. In general, the porosity of CNFs and their defectiveness is higher than that of MWCNTs [156,157], which can enhance their sensing characteristics. However, there are very few investigations devoted to CNF-based ammonia gas sensors. Monereo et al. [154] investigated a flexible sensor based on CNFs using inkjet-printed electrodes on a Kapton substrate (Figure 5). Two types of carbon nanofibers with different degrees of graphitization (≈70% and ≈100%) were used.

Claramunt et al. [158] investigated CNFs decorated with metal nanoparticles on a Kapton substrate and a polyimide flexible substrate with interdigitated electrodes and a heater at the back end (Figure 6). It was found that this decoration affects the response of CNF-based sensors differently. For example, decoration with Pd enhances the sensor response, whereas Au diminishes it. The authors proposed two sensing mechanisms that can take place in hybrid materials with decorated metal nanoparticles. Firstly, there is the adsorption of ammonia directly by the defects of the CNF sidewall. The second mechanism includes the adsorption of molecules by metal nanoparticles, which leads to a change in the charge transfer between CNFs and nanoparticles.

It is worth noting that the most popular type of modification of CNTs and CNFs for the enhancement of gas sensing is chemical treatment. The effect of other types of modifications, e.g., heat treatment [159,160], for ammonia detection is poorly studied.

A comparison of the characteristics of ammonia gas sensors based on SWCNTs, MWCNTs, CNFs, and their hybrids is presented in Table 1.

## 4. Gas Sensors Based on Graphene, Graphene Oxide, and Related Materials

Generally, graphene oxide (GO), and graphite oxide are insulators, limiting their application as gas sensors, but there are some publications that have reported their successful use as active materials. Bannov et al. [92] created a room temperature chemiresistive gas sensor for ammonia detection based on pristine graphite oxide. It was reported that the sensor possessed a good response to NH_3_ in dry synthetic air (ΔR/R_0_ was 2.5% for 100 ppm at 3% RH) (Figure 7a). As can be seen in Figure 7b, the response of graphite oxide to ammonia strongly depends on the relative humidity and changes slightly in the range of 3–27%, after which it grows considerably. The highest response (22.2% for 100 ppm NH_3_) was reached at a 65% relative humidity level.

It can be noted that the majority of articles on GO-based ammonia sensors are devoted to Hummers’ GO, whereas less attention is paid to GO synthesized by Hoffman, Brodie, Tour, etc. In particular, the composition of surface functional groups in GO obtained by different methods and its effect on sensor characteristics is of interest.

Generally, the sensing characteristics of graphene oxide and graphite oxide are relatively low; therefore, various approaches are used to enhance them. For example, Zhu et al. [168] created a film of graphene oxide to coat a laser-textured silicon substrate. The application of such approach made it possible to obtain a response ranging from ≈0.94 to 2 ppm NH_3_ and ≈0.7 to 100 ppm NH_3_. It was noted that the textured substrate provides more efficient adsorption sites for ammonia. The authors reported that the textured substrate could be useful for other graphene-based materials used in the field of sensors.

The functionalization of graphene oxide is widely used to enhance the sensitivity and selectivity toward ammonia. Kumar et al. [169] investigated a room temperature gas sensor based on 2-amino pyridine-functionalized graphene oxide. Sensors were prepared by the drop-casting technique on Si/SiO_2_ substrates with Al contacts. The sensor exhibited a response of 8.5% and 55.6% at 50 ppm and 900 ppm, respectively (Figure 8a). The high response of the sensor based on the 2-amino pyridine-functionalized GO was explained by the physical adsorption of NH_3_ on the surface of GO and the formation of hydrogen bonds (NH–O) with oxygen-containing functional groups of GO (Figure 8b). The sensor showed good selectivity and long-term stability in its response (Figure 8c,d).

Kumar et al. [88] fabricated an ammonia sensor based on meta toluic acid-functionalized GO on an SiO_2_/Si substrate using the Langmuir–Blodgett method, which had the highest response of 12.2% to 100 ppm NH_3_. The esterification reaction between GO and meta toluic acid led to the formation of ester groups that led to the realization of one of the possible mechanisms of ammonia gas sensing, namely the interaction of ammonia with oxygen atoms of ester groups through hydrogen bonding (H_2_NH·O). The authors of [170] also aimed for the formation of ester groups between hydroxyls of GO and carboxylic groups of benzoic acid in order to activate the same sensing mechanism. Aryl fluoride functionalization of GO using a “click” reaction was carried out in [171]. Enhanced ammonia sensing was shown by 2,3-difluoro and 2,3,4-trifluoro-substituted aryl propargyl ether-functionalized GO with a response/recovery of 63%/90% and 60%/100% at 20 ppm, respectively. The computational results confirmed that the adsorption energy was increased on functionalized graphene oxide (−1.74 eV for pristine GO, and for functionalized GO, it ranged from −2.14 to −2.89 eV). The key role of F atoms is their contribution, by means of p-electrons, to the adsorption of ammonia. The fluorination of GO can also be considered as one of the ways to improve its sensing behavior. In [172], the sensor response of chemically fluorinated GO was approximately 20 times higher than that of rGO.

The decoration of graphene oxide and reduced graphene oxide with nanoparticles makes it possible to improve the sensitivity of room temperature chemiresistive gas sensors. In [173], rGO/Ag nanoparticle nanocomposites on Cr/Ag interdigitate electrodes (fabricated by lithography) with a 100 ppt limit of ammonia detection were created. Surface plasmonic resonance when exposed to visible light (blue LED, 10 mW/cm^2^) in Ag nanoparticles made it possible to obtain ≈1.7 times better sensitivity for ammonia detection (Figure 9). Good selectivity to ammonia compared to VOC gases made the sensor a good candidate for breath analysis.

Ly et al. [174] fabricated thin films based on GO and ligand-capped Au nanoparticles using Langmuir–Schaefer and Langmuir–Blodgett techniques. The enhanced sensitivity of the films was mainly attributed to the Au nanoparticles, which act as active catalysts, providing spillover zones and accelerating the diffusion from active sites to inactive sites.

The textural characteristics of graphite oxides and their hybrids (used as ammonia gas sensors) are not often studied. However, Travlou et al. [71] created a set of hybrids based on Cu–benzene tricarboxylic (BTC) organic ligand metal organic frameworks (MOF) and graphite oxide. The surface area of the latter was 9.2 m^2^/g, whereas it was 53–916 m^2^/g for the hybrids. The ability of the amorphous MOF phase to absorb ammonia, in combination of the conductive graphene phase, made it possible to create reversible sensors. The adsorption mechanism of ammonia in this hybrid material is relatively complex, including ammonia complexation to metal sites, acid–base interactions with the carboxylic groups of the ligands, and direct interactions with the graphene phases, interactions with the graphene phases through dispersive forces and weak reactions between ammonia and BTC acid.

A comparison of the characteristics of ammonia gas sensors based on graphene, GO, graphite oxide, and related materials is presented in Table 2.

The reduction of graphene oxide or graphite oxide made it possible to obtain porous materials with higher specific surface areas than are desirable for gas sensors. At the same time, any reduction led to a reduction in the concentration of oxygen-containing functional groups, obstructing the capture of ammonia molecules. Usually, GO in gas sensors can be considered as a p-type semiconductor, the resistance of which grows upon contact with NH_3_, as well as rGO. However, there are some exceptions. For example, Lu et al. [95] observed the opposite change in the resistance of a GO sensor after 2 months compared to the as-reduced sensor. A simplified model, simulating the abnormal behavior of a current passing through the material, was proposed. It was supposed that there are two mechanisms of change in the conductivity of a sensing material, one of which induces the increase in conductivity and the other that has the opposite effect. The real behavior of the sensor is a combination of these two mechanisms.

Finally, it can be noted that active materials based on CNTs and CNFs are mainly directed toward the use of pristine untreated materials, whereas the issue of the application of doped materials is of interest and requires extended investigations in relation to ammonia detection.

Some publications reported on the fact that GO sheets have no response to ammonia, showing an extremely low change in the electrical properties of graphene oxide in the non-reduced state. Therefore, the reduction of GO is frequently used to enhance the sensing behavior of active material. In [95], the authors found that the heating of graphene oxide at 100 °C for 1 h made it responsive to gases such as NO_2_ and NH_3_. This fact is explained by the partial reduction of GO as a result of heat treatment and the creation of defects facilitating active sites for the adsorption of gas.

The curvature of rGO nanosheets plays an important role in ammonia gas sensing. Zhang et al. [175] obtained highly wrinkled rGO nanosheets from ball-milled graphite powder using chemical exfoliation. It was found that the wrinkled structure of rGO nanosheets provides a free space for the diffusion of gas and intensive changes in the adsorption energy compared to flat nanosheets (Figure 10).

Ni sulfate-doped rGO ammonia sensors were investigated using interdigitated electrodes in [183]. rGO in hybrid nanostructures can be considered as an important component, enhancing the detection of ammonia [33,72,94,120]. In [184], a hybrid nanocomposite consisting of rGO and In_2_O_3_ ceramic nanofibers was obtained. A synergetic effect between ceramic nanofibers and rGO was achieved, indicating a 10-fold faster response than the individual components and a low detection limit (44 ppb). A similar synergetic effect was achieved in [185] for a quantum dot/rGO composite. In [176], the authors reported on the enhancement of the properties of a hybrid material when adding WS_2_ to rGO, confirming the strong adsorption ability of the first and the introduction of acid centers by WS_2_. Sakthivel et al. [186] created an rGO–CuO nanocomposite sensor with a response (R/R_0_) of 13 at 30 °C and 30 at 300 °C, which is higher than that of the pristine CuO device. It was found that the nanocomposite had a three times higher surface area than pristine CuO, indicating that the flower-like CuO on rGO provides more adsorption sites. Nanoparticles of CuO had a valuable effect on the sensitivity of the PANI/CuO 3D-N-doped graphene nanocomposite described in [132].

Particular attention was paid to sensors based on reduced graphene oxide (rGO) and PANI [119]. Lee et al. [177] found that the response of PANI/rGO ammonia gas sensors increased when decreasing the PANI/rGO ratio. The optimal value of the ratio was 125. It was assumed that rGO nanosheets with a large surface area ensured the homogeneous distribution of PANI nanospheres. The response of the PANI/rGO sensitive membrane was higher than for individual components (0.5%, 8.3%, and 13.0% for rGO, PANI, and PANI/rGO under 15 ppm NH_3_, respectively). N-rGO/PANI nanocomposites have been shown to possess a response higher than that of rGO/PANI and GO/PANI sensors as a result of the p–n heterojunction [187].

Despite the creation of hybrids based on rGO and dissimilar materials, there are also hybrids based on rGO and other carbon materials. For example, a hybrid gas sensor for ammonia detection based on rGO and graphene was created by Wang et al. [188]. Here, rGO plays the role of the adsorption material to capture the ammonia molecules. Huang et al. [189] developed a 3D framework of rGO using silica as a template, showing an enhanced response of 31.5% to 50 ppm NH_3_, which was significantly higher than the 2D rGO network with a response of 1.5%.

Graphene is considered to be more sensitive to ammonia, since it has a 2D structure with an extremely high specific surface area (2600 m^2^/g) and a high signal-to-noise ratio [190]. There is a difference between the application of graphene and rGO in gas sensors. The first has a higher surface area and possesses higher conductivity compared to rGO, which has some oxygen-containing functional groups as a result of its reduction [191]. The role of defects in graphene for the detection of ammonia play an important role (a theoretical analysis of defective graphene for ammonia gas sensing applications was carried out in [192]). The adsorption energy of ammonia by various types of graphene was −0.11 eV (pristine graphene), −0.12 eV (N-doped graphene), and −0.24 eV (defective graphene) according to calculations carried out in [193].

Sensors based on graphene are manly created in the form of hybrids. In [181], graphene film coated with SnO was used for the detection of ammonia. The response was 10% to 20 ppm, and the sensor showed long-term stability (over 8 months). It was noted that no heating was required for the recovery of the sensor. The fast operation of the sensor was attributed to the Sn attachment with a single oxygen molecule linked to graphene. Kodu et al. [182] investigated two types of graphene (CVD, transferred onto oxidized silicon, and epitaxial, grown on SiC) functionalized with sputtered V_2_O_5_. SiC/epitaxial graphene showed a response one order of magnitude higher (295% to 100 ppm NH_3_) than functionalized CVD graphene (31% to 100 ppm NH_3_). This effect was explained by the smaller initial free charge carrier doping in epitaxial graphene. A novel system for the testing of graphene sensors functionalized with Co(tpfpp)ClO_4_ was created in [194].

Heteroatom-doped graphenes are excellent candidates for gas sensors [195]. The investigation of other types of graphenes showed interesting results and reflect the potential of these materials for ammonia detection. Fluorinated graphene has been extensively studied for ammonia detection [100,180,196], since it contains electronegative F atoms, and their interaction with hydrogen is one of the strongest interactions that occur in hydrogen bonding, therefore enhancing the adsorption of gases [197]. The absorption energy of fluorographene was significantly higher than that of graphene (0.137 and 0.026 eV, respectively) [197]. In [99], the films of fluorinated and oxifluorinated graphene-based room temperature sensors were investigated. It was shown that the films possessed the same sensitivity to ammonia, but the first one showed a better recovery (the recovery time was 1100 s and 2500 s, respectively). Experiments carried out in [198] showed that boron-doped graphenes are of interest for gas sensing. The response of B-doped graphene was 8.92% to 32 ppm NH_3_ when compared to pristine graphene (2.64%) [199]. N-doped graphene is also a good potential candidate for the detection of ammonia [132].

## 5. Effect of Operating Temperature and Humidity on Sensing Characteristics

The issue of operating temperature is very crucial for ammonia gas sensors. Many ammonia gas sensors operate at room temperature, and research on the effect of the operating temperature on the sensing characteristics of MWCNTs was carried out in [86]. It was found that sensors exhibited a higher response to ammonia at 200 °C than that at room temperature. This effect was explained by the thermodynamics of absorption, indicating that this process is exotheric and any increase in temperature shifts the equilibrium to the desorption side. The incomplete desorption of ammonia on MWCNTs was also reported, showing the more complete recovery of the sensors at 200 °C compared to room temperature (Figure 11).

It can be noted that the difference between the responses of the sensor became higher at a concentration of NH_3_ below 500 ppm; i.e., there is no difference between the response at relatively high concentrations of ammonia. The decrease in the sensor response was also reported in [67] for rGO gas sensors and [176], where the response of an rGO/WS_2_ hybrid sensor decreased from 93% to 27% (100 ppm) (Figure 12). For some hybrid materials, the dependence of the response on the temperature is relatively complex. For example, it was reported in [158] that the response increases up to 70 °C and then decreases at higher temperatures. The maximum response is attributed to the temperature where the adsorption and desorption are balanced. The optimum temperature for CNF-based sensors was 110–120 °C at 50% RH. It can be noted that the response time becomes faster at a higher temperature; i.e., saturation occurs more quickly [200].

## 6. Humidity and Its Impact on Sensing Characteristics

In the majority of articles published on the impact of relative humidity on the ammonia sensing properties of carbon nanotubes [24], carbon nanofibers [153], graphite oxide, and graphene oxide [92], it was reported that the sensor response increases with an increasing RH.

Generally, ammonia dissolves in water adsorbed on the surface of carbon nanomaterials [24,96,201], according to the reaction shown below:NH_3_ + H_2_O = NH_4_^+^ + OH^−^.

The base ionization constant K_b_ for this reaction is 1.8 × 10^−5^ (25 °C) [24].

The increased response of gas sensors in wet air depends on the features of the material as well as the method of its functionalization. The mechanism of change in the sensor response becomes complex when ammonia comes into contact with hybrids based on carbon nanomaterials in wet air. In [24], it was found that COOH functionalization of SWCNTs led to an increase in the interaction of ammonia molecules with functionalized CNT surfaces. Chen et al. [122] determined the effect of RH on the response of SWCNTs–OH/PANI composites, and this increased with an increasing RH below 85%. A further increase in humidity led to a decrease in response, which might be attributed to the exchange of protons between ammonia and PANI-based materials in the presence of molecules of water. The scheme of the process, illustrating that H^+^ and OH^−^ ions can interact with amine groups in PANI, is shown below [202,203]:NH_2_^+^ + H_2_O → NH + H_3_O^+^OH^−^ + H_2_O → O_2_ + H_3_O^+^.

Water molecules absorbed by PANI chains act as donors of protons, inducing a reduction in resistance. A similar effect in terms of a decrease in sensor response RH increases was also reported in [204] for SWCNTs/Pyrene 3D film, showing a 1.6–2-fold decrease in response upon an increase in RH from 40% to 70%. This is caused by the competitive sorption of ammonia and water molecules on the surface of the active layer. One of the side effects of the detection of gas in humid air is the relatively high affinity of carbon nanomaterials to the adsorption of water vapor. Therefore, these materials can also be utilized in humidity sensors [205].

Some of the sensors show almost no change in the response and recovery time upon an increase in RH, e.g., graphene/SnO hybrid sensors [181], but this effect must be thoroughly investigated.

## 7. Recovery of Sensors

Mainly, sensors based on carbon nanomaterials require recovery. This recovery can be carried out using heating [87,153,162], an increase in gas flow [40], a strong electric field [206], or irradiation by infrared light [207]. In [207], a sensor based on graphene decorated with Au nanoparticles was illuminated with infrared radiation (≈2 µW/mm^2^), which made it possible to recover more than 90% of the original baseline level. This effect is caused by the strong absorption of infrared radiation by the graphene surface.

Visible light and UV have an effect on the response and repeatability of the sensor response. According to [154], CNFs were more sensitive to visible light than to UV. It has been shown that CNFs with different graphitization degrees showed different behaviors under UV radiation in terms of their changing resistance. This fact indicates that the effect of visible light on the performance of ammonia gas sensors cannot be ignored. It has been shown that the response of ammonia gas sensors based on CNFs changes under UV light [155]. The accumulation of a response with no recovery was found without UV radiation. Complete desorption was achieved via UV illumination. The response decreased when the power of lamp increased, whereas the response time and recovery time decreased (5.6 s and 24 s for 40%; 2.2 s and 19.4 s for 80%, respectively) [155].

Heating is one of the most frequently used approaches for sensor recovery, and sensors can be operated using external heaters [139,208] or self-heating [209,210]. The thermal impact induces the desorption of NH_3_, and the temperature determines the energy required to overcome the binding of ammonia molecules with the surface of carbon nanomaterials [139]. Quang et al. [87] performed the recovery of SWCNT gas sensors using a temperature range of 60–100 °C (Figure 13a). A reproducible complete recovery of ammonia molecules was achieved at 80 °C for 5 min (the flow rate of the carrier gas, nitrogen, was 1000 sccm). In [153], heating (100 °C, 3 min) was used for the 100% recovery of sensors based on nanofibrous carbon. In [162], the protocol of heat treatment for SWCNT/Al_2_O_3_ ammonia gas sensors was 200 °C for 10 min in a nitrogen flow. Sharma et al. [139] showed that 150 °C provides sufficient energy to carry out the recovery of ammonia from the surface of MWCNTs in MWCNTs/Al_2_O_3_ composite-based sensors. Monereo et al. [104] found that the self-heating of randomly oriented fishbone-like CNFs made it possible to reach a temperature of up to 250 °C with a consumption of tens of µW. The regimes of self-heating of CNF-based gas sensors via pulsed heating and various levels of RH, as well as their impact on response, were studied in [211].

In the thermal recovery of ammonia gas sensors based on graphene oxide and graphite oxide, the temperature of the recovery plays an important role. This is caused by the temperature during the reduction of GO, which is unique for various types of materials and depends on the C:O ratio, synthesis technique, content of functional groups, and other characteristics. Therefore, the temperature of the recovery cannot be increased to a higher value than that which triggered the onset of the reduction of graphene oxide or graphite oxide [212]. Nonetheless, this problem is still poorly considered in the article devoted to sensors based on GO.

DC bias also has an influence on the recovery of sensors. In [134], the simultaneous application of a DC-biased current and heating was proposed to achieve the complete recovery of conducting polymers/MWCNTs sensors (Figure 13). The best recovery (98.5%) was shown at 150 °C and 3 mA. Mishra et al. [206] observed the poor recovery of MWCNTs grown in SiO_2_/Si substrate ammonia gas sensors, and a DC electric field was applied across the electrodes of sensors. The enhanced recovery was explained by the supply of sufficient energy in order to achieve the jump of electrons through the defect sites (Poole–Frenkel effect), inducing the desorption of gas. Recovery increased when the current increased from 1 to 13 mA.

Despite the fact that heating is widely used for the recovery of ammonia gas sensors, the data on the change in the concentration of ammonia on the surface of carbon materials during heating are almost absent. In addition, the criteria for the selection of the heating time and temperature are not clearly formulated, and simulations should be carried out.

## 8. Methods of Deposition of Carbon Nanomaterials on the Substrate

Considering the methods of deposition of carbon nanomaterials on the substrate, it can be noted that most of them use spin coating [73,200], drop casting [213,214], spray coating [92], etc. However, there are some special methods that can be used to obtain sensors. For example, Mishra et al. [27] used a gel-casting technique for the preparation of ceramic/SWCNT films, which were prepared via a special motor-driven machine fitted with a doctor blade, where an SWCNT powder was dispersed in sol–gel alumina solution. The solution was poured and rolled out on Mylar tape while controlling the thickness of the film. The nanocomposite film showed a response of 15% to 1 ppm NH_3_.

Generally, the as-received CNTs are deposited on the SiO_2_/Si substrate using a material dispersed in solvent. The technique of the direct growth of carbon nanomaterials on an SiO_2_/Si substrate is not particularly common. In [142], catalytic Fe nanoparticles and MWCNTs were deposited directly onto an SiO_2_/Si substrate in a plasma reactor with a microwave dual-flow nozzle electrode.

In some cases, the creation of flexible films for ammonia gas sensing requires the proper transfer of the product deposited on the substrate onto a flexible substrate (e.g., PET [166,215], PI [216], cellulose [217], PS [218,219], PANI [215], etc.) or the creation of mats [219]. In [166], vertically aligned MWCNTs were transferred onto a PET substrate by means of hot pressing. The CNTs were primarily deposited onto the SiO_2_ substrate using cold wall thermal chemical vapor deposition. In [217], flexible SWCNT–paper structures for ammonia detection were prepared by drop coating. It was reported that cellulose fibers can also react with ammonia according to the following reaction:OH + NH_3_ → NH_2_ + H_2_O.

However, this reaction cannot change the resistance of cellulose significantly. In any case, inter-tube charge transport between cellulose fibers does occur, to which CNTs can make a contribution. Therefore, this effect of ammonia interaction with cellulose may change the resistance of sensors. Moreover, four sensor substrates were tested (glass, Teflon, paper, and a composite), and the highest response was achieved for SWCNTs on a glass substrate.

## 9. Challenges

The following challenges for the room temperature gas sensors for ammonia detection can be highlighted below:Effect of relative humidity. The sensing characteristics depend on the humidity of air and mostly, the response to ammonia grows when increasing RH. The main task is to engineer such a surface chemistry that diminishes the effect of humidity, making the sensitivity, response/recovery time, and other important characteristics unaffected.Selectivity. The problem of enhancing the selectivity of carbon nanomaterials is still poorly described. There is not enough data published to know the factors affecting selectivity. Of course, doping and additional functionalization of CNTs/CNFs, graphenes, and related materials will enhance selectivity, but common relationships between them must be found.Scale-up. There are a lot of various carbons used for the detection of ammonia, and nowadays, the hybrids and composites are of particular interest, but the possibility to scale up the technological processes must be carefully checked on the basis of calculations. Involving chemical engineering approaches into the study of sensors is extremely necessary. The preparation techniques (especially, spin coating, spray coating, etc.) should be evaluated for their repeatability and possibility to create the large arrays of devices.Green chemistry and gas sensors. Many synthesis techniques (e.g., chemical vapor deposition, plasma enhanced chemical vapor deposition) may harm the environment. Therefore, the application of green chemistry approaches for the synthesis of active layers of graphenes and relative materials to be considered for the detection of ammonia.

## 10. Conclusions

In the present work, a review of ammonia gas sensors based on carbon nanomaterials has been presented. The features of the change in response, the effect of relative humidity, the problem of the recovery of sensors based on various carbon materials (carbon nanotubes, carbon nanofibers, graphene oxide, graphite oxide, graphene, etc.) and their hybrids have also been described. Some studies showed the different behaviors of various carbon materials in terms of ammonia detection. From the review conducted, important relations between the properties of carbon nanomaterials and their sensing behaviors can be made.

## Figures and Tables

**Figure 1 micromachines-12-00186-f001:**
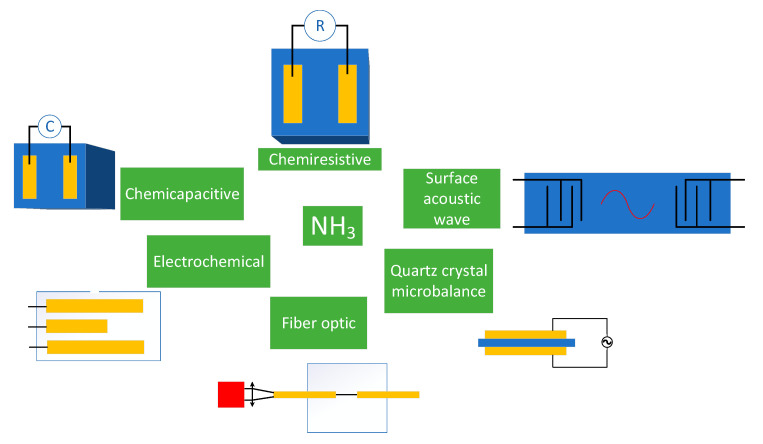
Different types of ammonia gas sensors.

**Figure 2 micromachines-12-00186-f002:**
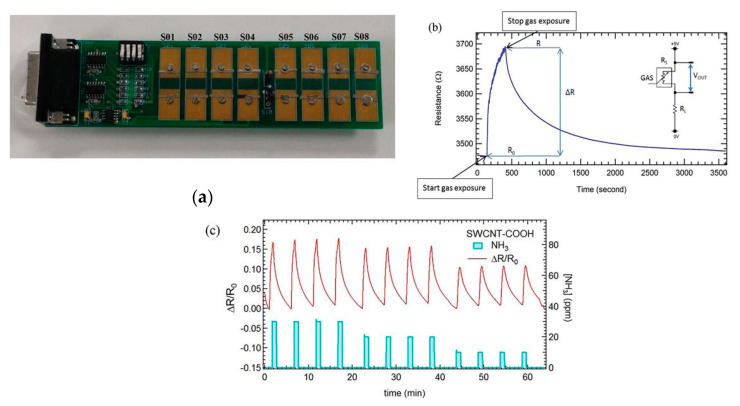
(**a**) Sensor array board of eight sensors based on single-walled carbon nanotubes (SWCNTs) functionalized with various organic molecules (S01:DNA, S02: PANI, S03: TCTA, S04: TAPC, S05: PTCDA, S06: ex-4T-Hex, S07: COOH, SO8: CNT); (**b**) Change of resistance of p-doped CNTs to ammonia; (**c**) Sensors response ΔR/R_0_ for CNT-COOH sensor followed by a sequence of four exposures to 30, 20, and 10 ppm NH_3_ [23]. With the permission of John Wiley and Sons.

**Figure 3 micromachines-12-00186-f003:**
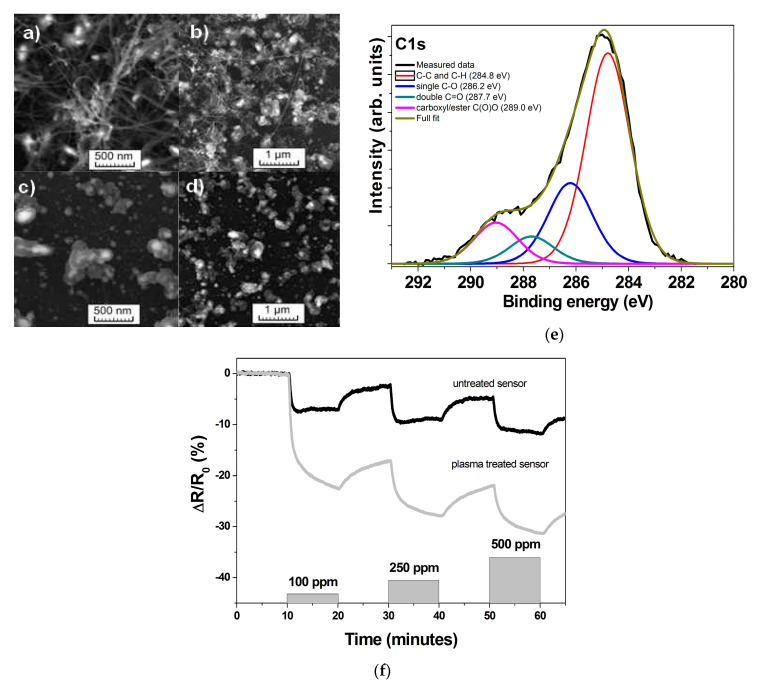
(**a**–**d**) SEM images of multi-walled carbon nanotubes (MWCNTs) before (**a**,**b**) and after treatment (**c**,**d**); (**e**) XPS spectrum of carbon nanomaterial after plasma functionalization; (**f**) Response of plasma functionalized sensor compared to untreated one [142]. With the permission of IEEE.

**Figure 4 micromachines-12-00186-f004:**
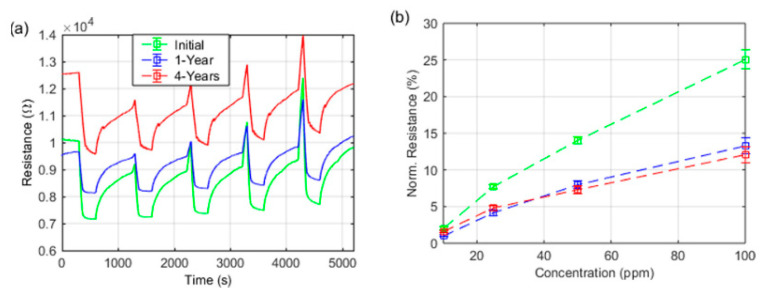
(**a**) Response to NH_3_ vs. time after 1 day, 1 year, and 4 years from fabrication; (**b**) Calibration curves for NH_3_ after 1 day, 1 year, and 4 years from fabrication [152]. IOP Publishing. Reproduced with permission. All rights reserved.

**Figure 5 micromachines-12-00186-f005:**
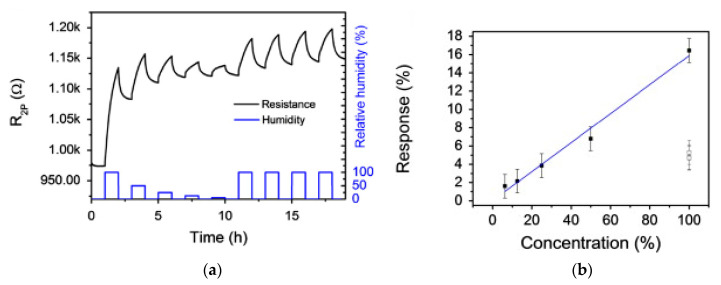
(**a**) Change of resistance depending on humidity of spray deposited carbon nanofibers (CNFs); (**b**) Effect of humidity on the sensor response [154]. With the permission of Elsevier.

**Figure 6 micromachines-12-00186-f006:**
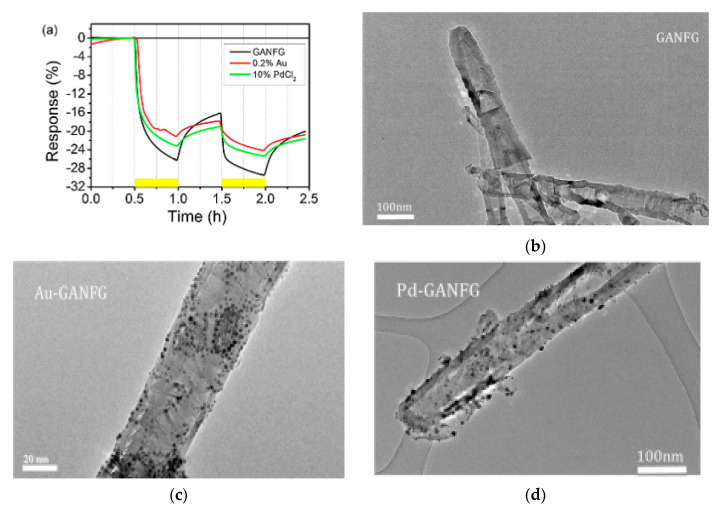
(**a**) Response of the sensors in the presence of 500 ppm NH_3_ at 22 °C; TEM images of (**b**) raw carbon nanofibers (sample code was GANFG), and decorated (**c**) Au-GANFG and (**d**) Pd-GANFG samples [158]. With the permission of Elsevier.

**Figure 7 micromachines-12-00186-f007:**
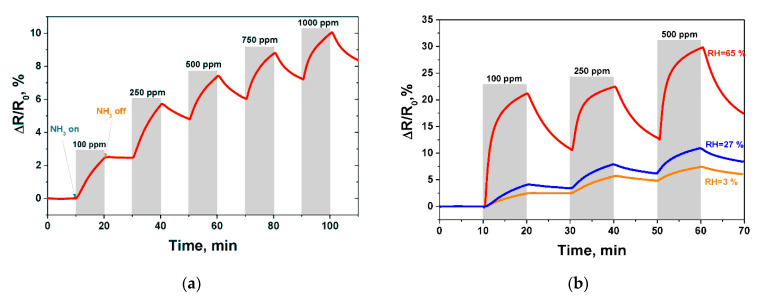
Chemiresistive sensor operating at room temperature: (**a**) Response to ammonia in the range of concentrations from 100 ppm to 1000 ppm (RH = 3%); (**b**) Response to ammonia at various relative humidity levels [92]. With the permission of MDPI.

**Figure 8 micromachines-12-00186-f008:**
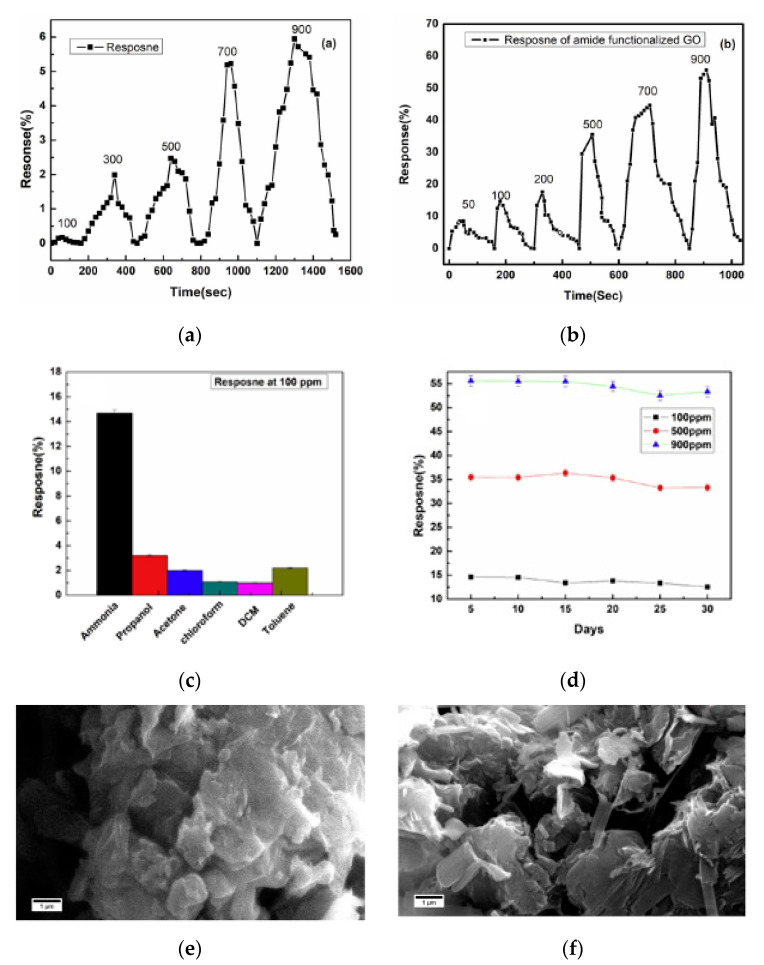
Response curves of (**a**) graphene oxide (GO) and (**b**) Amide functionalized GO; (**c**) Selectivity (concentration of gases was 100 ppm) and (**d**) Long-term response change of sensor based on functionalized GO; (**e**) SEM images of GO and (**f**) amide functionalized GO [169]. With the permission of Elsevier.

**Figure 9 micromachines-12-00186-f009:**
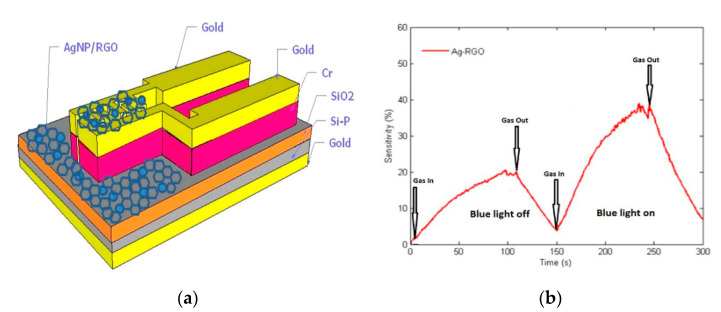
(**a**) Scheme of sensing layers of AgNPs/reduced graphene oxide (rGO) sensor; (**b**) AgNPs/rGO response vs. time at 20 ppm NH_3_ with and without irradiation by visible light (10 mW/cm^−2^) [173]. With the permission of IOP.

**Figure 10 micromachines-12-00186-f010:**
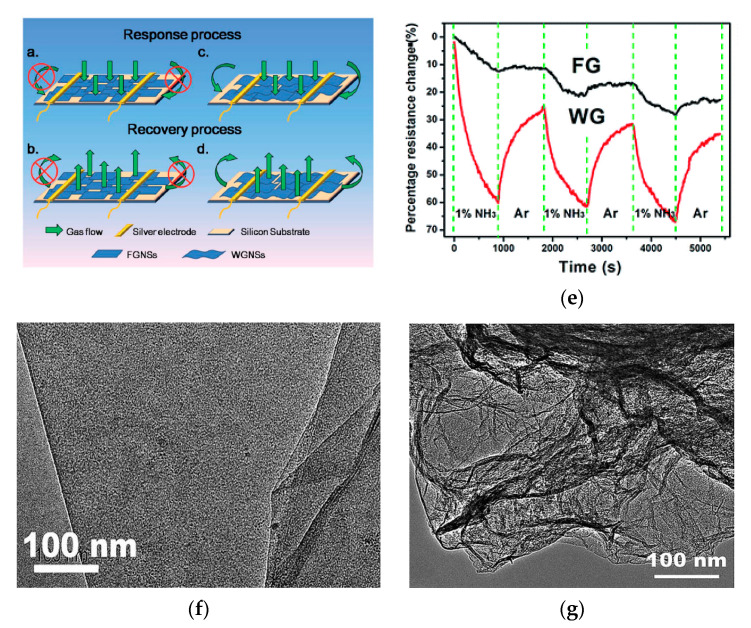
(**a**) Sensing of ammonia on flat rGO (FGNSs) and wrinkled rGO (WGNSs) nanosheets; (**b**) sensing properties of wrinkled and flat rGO nanosheets in 1% NH_3_; (**c**) TEM image of flat rGO; (**d**) TEM image of wrinkled rGO [175]. Reproduced by permission of The Royal Society of Chemistry.

**Figure 11 micromachines-12-00186-f011:**
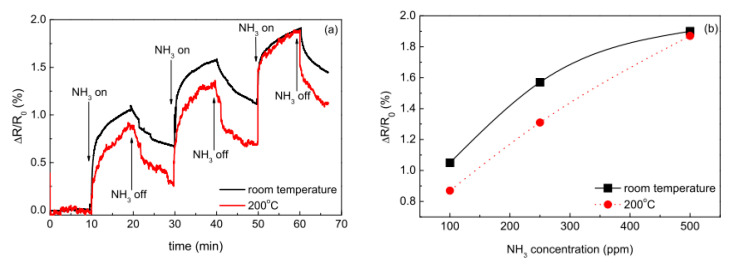
(**a**) ΔR/R_0_ curves in time for MWCNT sensor at room temperature and at 200 °C (NH_3_ concentrations were 100, 250, and 500 ppm, respectively); (**b**) ΔR/R_0_ of the sensor depending on the concentration of ammonia [86]. With the permission of MDPI.

**Figure 12 micromachines-12-00186-f012:**
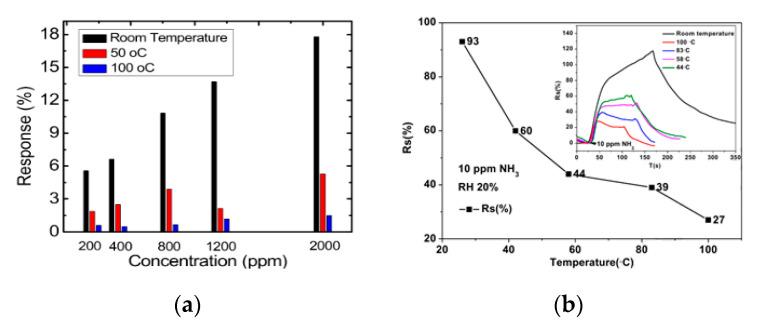
(**a**) Effect of operation temperature on the response of chemically reduced GO [67]. Reprinted (adapted) with permission from (Ghosh, R.; Midya, A.; Santra, S.; Ray, S. K.; Guha, P. K. Chemically reduced graphene oxide for ammonia detection at room temperature. ACS Appl. Mater. Interfaces **2013**, 5, 7599–7603.). Copyright (2021) American Chemical Society. (**b**) Variation of response of rGO/WS_2_ hybrid sensor at 10 ppm NH_3_ depending on the temperature [176]. With the permission of Elsevier.

**Figure 13 micromachines-12-00186-f013:**
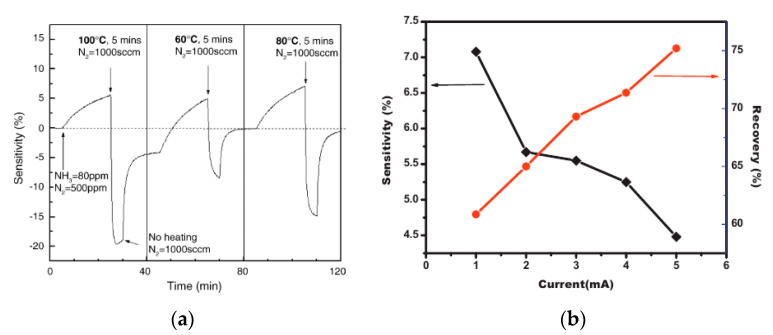
(**a**) Recovery of the SWCNT sensor at various conditions of gas desorption. In each period, NH_3_ is turned on at 80 ppm in 500 sccm N_2_ flux for 20 min. The desorption is proceeded at 1000 sccm N_2_ flux, begins by heating for 5 min at 60–100 °C; then, there was no heating for 15 min [87]. (**b**) Dependence of sensitivity and recovery percentage on the DC bias current (at 130 °C) [134]. With the permission of Elsevier.

**Table 1 micromachines-12-00186-t001:** Characteristics of ammonia gas sensors based on SWCNTs and MWCNTs.

Material of Sensor	Operating Temperature and RH	Concentration	Response (ΔR/R_0_)	Response Time/Recovery Time	Ref.	Substrate (Method of Deposition)
CTAB (cetyltrimethyl ammonium bromide) functionalized SWCNTs	Room temperature, 40% RH	10 ppm	≈6	16.7 ± 0.2 s/127.9 ± 2.1 s	[24]	Roll coating (flexible sensor)
Functionalized SWCNTs	40 °C	8 ppm	5.8%	3 min/7 min	[161]	Glass (spin coating)
COOH-functionalized SWCNTs	Room temperature	300 ppm	30%	400–300 s (response time)	[123]	n/a (flexible sensor)
SWCNT/Al_2_O_3_ film	Room temperature	1 ppm	15%	n/a	[162]	Gel-cast technique
SWCNTs-OH/PANI	25 ± 2 °C, 62 ± 3% RH	100 ppm	14.91%	81 s/149 s	[122]	Si (drop coating)
SWCNT/ZnPc-CF (fluoroalkyl-substituted zinc(II) phthalocyanines)	Room temperature	10 ppm	≈1.7·10^−2^	30 s/85 s	[163]	Glass (spin coating)
MWCNTs-OH	Room temperature	14 ppm	2.5%	n/a	[141]	Filter paper (filtration from suspension)
PANI-coated MWCNTs	Room temperature	5–150 ppm	≈0.1–1.5	90 s/100 s (5 ppm)	[117]	Spin coating
Polythiophene/MWCNTs	Room temperature	0.1 ppm	27.66%	n/a	[137]	In situ chemical oxidative method
MWCNTs/Al_2_O_3_	-	5 ppm	≈0.7%	10 min/n/a	[139]	Sol–gel method
Co/MWCNTs	Room temperature	7 ppm	≈1.0%	30 s/200–500 s	[164]	Alumina (screen printing)
MWCNTs/WO_3_ nanobricks	Room temperature	10 ppm	6.8%	n/a	[146]	SiO_2_/Si (drop casting)
MoS_2_/MWCNTs	Room temperature (27 ± 3 °C, 40% RH)	150 ppm	≈30%	400 s/280 s	[165]	Quartz
CNFs	Room temperature (25 °C, 100% RH)	1000 ppm	≈3.9%	n/a	[154]	Kapton (spray coating)
CNTs	Room temperature	50 ppm	≈1.0%	33 s/60 s	[166]	PET (hot pressing)
SWCNTs	Room temperature	50 ppm	≈12.5%	24.9 ± 3 s/ 323.9 ± 5 s	[167]	SiO_2_/Si (spray coating)
Aligned nitrogen-doped MWCNTs	Room temperature	1%	4.7%	2–3 s	[25]	Film sensor

**Table 2 micromachines-12-00186-t002:** Characteristics of ammonia gas sensors based on graphene, graphene oxide (GO), and related materials.

Material of Sensor	Operating Temperature and RH	Concentration	Response	Response Time/Recovery Time	Ref.	Substrate (Method of Deposition)
Pristine graphite oxide	Room temperature (25 ± 2 °C)	100–500 ppm	2.5–7.4% (RH 3%) 22.2–29.6 (RH 65%)	n/a	[92]	SiO_2_/Si (spray coating)
Wrinkled reduced graphene oxide	Room temperature	1%	35%	n/a	[175]	SiO_2_/Si (spin coating)
Cu-BTC/graphite oxide	Room temperature	100 ppm	1.7%	n/a	[71]	Alumina
2-amino pyridine functionalized GO	Room temperature (27 °C), 30 ± 4%	50–900 ppm	8.5–55.6%	20 s/120 s	[169]	SiO_2_/Si (drop casting)
GO functionalized with meta toluic acid	Room temperature	100–2000 ppm	12–22.7%	55 s/80 s	[88]	SiO_2_/Si (Langmuir–Blodgett)
rGO/WS_2_	33.5 °C, 20% RH	10–50 ppm	121–256%	60 s/300 s	[176]	Alumina (drop casting)
rGO	Room temperature	1%	30%	3 min/above 50 h	[95]	SiO_2_/Si (drop casting)
cpoPcCo/rGO	Room temperature (29 ± 0.5 °C)	100 ppm	42.5%	2.5 s/45 s	[93]	Drop casting
PANI/rGO	Room temperature (25 °C)	50 ppm	59.2%	n/a	[119]	Drop casting
PANI/rGO	Room temperature (27 °C)	15 ppm	13.0%	n/a/22.1 min	[177]	Glass (spin coating)
GO/Au nanoparticles	Room temperature (25 ± 2 °C), 45 ± 5% RH	70 ppm	9.8%	n/a	[178]	Si/SiO_2_ (Langmuir–Blodgett, Langmuir–Schaefer)
Aryl fluoride functionalized GO	Room temperature	20 ppm	60%	78 s/260 s	[171]	Glass (drop casting)
GO functionalized by para chloro benzoic acid	Room temperature	100 ppm, 1200 ppm	6.7%, 18.9%	80 s/115 s	[170]	Langmuir–Blodgett
PANI/GO/PANI/ZnO	Room temperature (25 ± 2 °C), 65% RH	100 ppm	38.31%	30 s	[72]	Quartz (layer-by-layer deposition)
Hole-matrixed carbonylated graphene	Room temperature (32 ± 3 °C), 28% RH	50 ppm	14%	13 min	[179]	n/a
CdTe/aerographite	Room temperature (30% RH)	200 ppm	153%	21 s/72 s	[102]	n/a
Reduced C_2_F graphite	Room temperature	1%	6%	n/a	[180]	Copper
Graphene/SnO	Room temperature	3.5 ppm	5%	n/a	[181]	Insulator substrate (drop casting)
SiC/epitaxial graphene	Room temperature	8 ppm	50%	695 s/658 s	[182]	Si/SiO_2_ and SiC

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
