# Peer review of "Recent Advances in Ammonia Gas Sensors Based on Carbon Nanomaterials"

_micromachines, 2021, doi:10.3390/mi12020186_

Round 1

Reviewer 1 Report

The review script submitted by author related to development of ammonia gas sensors based on carbon nanomaterial’s is the good effort made by authors.

But frankly speaking there are so many review articles already published every years related to ammonia sensing, I do not find any novelty in this review script. And secondly how practically commercial utilization is possible using MWCNT and Graphene based ammonia sensor which are quiet expensive. And thirdly there are so many important biopolymer based ammonia sensors work with very low detection limit of ppb and ppq are reported in literatures but I cannot find those refrences available here in this script. This means that the authors have not thoroughly perform the literatures survey well. Apart from that few other suggestion are listed below for author attention.

Motivation of the script is lacking.

Author have mention “Rigoni et al. [6] achieved a more than 100-fold increase in ΔR/R0 for SWCNTs functionalized with CTAB (cetyltrimethyl ammonium bromide) surfactant compared toCOOH–SWCNTs (a fraction of the groups was 1–3 at.%) in the range of 10–30 ppm NH3.”. Provide discussion of the sensitivity and stability in present case. The detection limit range is also missing in it.

In page 4, author have mention ‘It is interesting that the functionalization of reduced graphene oxide does not lead to the improvement of the sensor response. The 162 creation of hybrid materials with phthalocyanine derivatives is successively achieved not only for SWCNTs, but also for reduced graphene oxide (rGO)’. Justify?

If we compare the length of script with that of references, It can be found that there are so many references (total 207) in the script, which need to be reduced.

Mechanism are lacking based on several carbon nanaomaterials. Several mechanisms should be focus and discussed in this script.

What are the future prospects of this work? And future challenges associated with such sensors.

Is author are talking about the ideal sensors in this script. I do not find the gas sensors reported in this script are ideal.

Reviewer 2 Report

The Review paper entitled "Recent advances in ammonia gas sensors based on carbon nanomaterials " from Bannov et. al has been submitted to MDPI Micromachines Journal. 

The importance of the topic is well defined and well documented by the appropriate use of references. 

However, there are several issues to be fixed by the authors before considering it accepted or suitable for its publication. 

  • English should be checked through the whole text. Very long and confusing phrases are used. Besides it is repetitive using words as: Gas Sensor, Devote and others. Avoid words like "cheap"(line 26) or phrasing like: We suppose (line 64)
  • The introduction section is confusing regarding its organization.  It starts talking about gas determination, then carbon nanotubes examples, then ammonia, then portable gas sensors, then ammonia once again... This should be re-organized in such a manner that is more attractive and easy for the reader. 
  • All figures are in very low resolutions. Besides, figure captions including references and copyright are separated. Labeling of the figures should be consistent and aligned. 
  • Figure 9: does not offer any useful information 
  • Insted of ppm or ppb, mg/L or μg/L may be used.
  • Tables: use limit of detection (and units) instead of concentration. Besides, linear response may be included if possible. 
  • The review is based in how carbon nanomaterials are useful for ammonia gas sensors, but there is a lack of electron microscopy images of those nanomaterials (only carbon nanotubes) Besides it would be interesting to include the importance of nanomaterials and why do they enhance sensing features, this in the introduction section. 
  • Small comments regarding environmental sensors and nanotechnology in the sensing field may be included in the introduction. Then, highlight why carbon (or nanocarbon) materials are selected. 
  • Section 4,5,6 are very general. These facts could have  been summarized and included as so, in the introductions, as they apply for all materials selected. 
  • Comparison with existing ammonia sensors or other analytical techniques may be taken into account 
  • Conclusion section is not well addressed. In a review paper, this should be like a perspective of the future of the field and the trends treated in the text. 
  • Conclusion section: graphene oxide is repeated. 
I advise major revisions to re consider further this paper before acceptance. 

Round 2

Reviewer 1 Report

The author was fail to address most of the comments raised in this script. Few of them are mention below for your attention.

  • But frankly speaking there are so many review articles already published every years related to ammonia sensing, I do not find any novelty in this review script. And secondly how practically commercial utilization is possible using MWCNT and Graphene based ammonia sensor which are quiet expensive. And thirdly there are so many important biopolymer based ammonia sensors work with very low detection limit of ppb and ppq are reported in literatures but I cannot find those refrences available here in this script. This means that the authors have not thoroughly perform the literatures survey well. Apart from that few other suggestion are listed below for author attention.

This comments was not address properly. Even check several articles are available such as Analyst, 2013,138, 7392-7399; Procedia Engineering 87 ( 2014 ) 716 – 719; Appl. Phys. Lett. 101, 053119 (2012); Nature Scientific Reports 3 (2082), DOI:10.1038/srep02082 (2013); ACS Sensors 1, 55-62. Review means thoroughly literature survey. So kindly find my advice and properly revised the script.

  • Mechanism are lacking based on several carbon nanaomaterials. Several mechanisms should be focus and discussed in this script.

Author response: The description of mechanism was added to Introduction. “Despite the difference of various carbon nanomaterials, the mechanism of change of resistance when exposing ammonia is not changed among them. The most of carbon nanomaterials undergo increase of resistance during adsorption of ammonia that is related to p-type of these materials (e.g., MWCNTs, CNFs). Ammonia donates electrons to active materials of sensor inducing a decrease of concentration of charge carriers (holes) [79,98]. The mechanism of change of hydrid sensor resistance is complex and depends on the type of hydrid. For example, in PPy-rGO sensor both materials behave as a sensing ones, and enhances the capture of ammonia molecule inducing increase of resistance [60]. The conjugated π-systems of hybrids zinc(II) phthalocyanine-SWCNTs (both components possess p-type conductivity) led to larger response [99].” There was no general mechanism for ammonia sensing on hybrids, therefore the only certain materials have been considered. In addition, the mechanisms are briefly discussed in sections 2-5

Did author understand what the question is all about is. I have mention to provide the detail mechanisms with different types of sensor which is used for ammonia sensing in present review script. Details along with schematic diagrams are needed. Not just the one paragraph explanation inside the introduction. Mechanism should be mention where author have discuss about several types of ammonia sensors.

So kindly seriously address the comments raised in this script.

Reviewer 2 Report

Suggested changes have been carried out. 

Some figures may be further treated to increase resolution and layout. 

I think this paper may be suitable for acceptance, depending on the final decision of academic editor. 

Author Response

The changes were made. The additional Figure was added. Unfortunately, the quality of Figures depends on the orinial files provided by the Publishers, therefore they are at the highest quality can be obtained. 

Round 3

Reviewer 1 Report

The author have satisfactory revised the script.